# Bean Sprouts, Lettuce, and Milk as Water Sources in *Tenebrio molitor* Larval Growth

**DOI:** 10.3390/ani14060895

**Published:** 2024-03-14

**Authors:** Seokhyun Lee, Andrew Wange Bugenyi, Hakkyo Lee, Jaeyoung Heo

**Affiliations:** 1Department of Animal Biotechnology, Jeonbuk National University, Jeonju-si 54896, Republic of Korea; sl144@jbnu.ac.kr (S.L.); breedlee@jbnu.ac.kr (H.L.); 2Department of Agricultural Convergence Technology, Jeonbuk National University, Jeonju-si 54896, Republic of Korea; 3National Agricultural Research Organization, Mbarara P.O. Box 389, Uganda; 4International Agricultural Development and Cooperation Center, Jeonbuk National University, Jeonju-si 54896, Republic of Korea

**Keywords:** insect production, *Tenebrio molitor*, larval growth, water feed level, water source

## Abstract

**Simple Summary:**

There is growing interest in insects as an alternative to animal proteins in the food and feed industries because of concerns about the unsustainability of existing production methods. Yellow mealworms (the larvae of the beetle *Tenebrio molitor*) are the most prominent insect being promoted for industrialization in this regard. Water requirements in mealworm farms rely on leafy vegetables (lettuce, napa cabbage, and white radish) or agricultural by-products, but their seasonality poses challenges in supply and storage. This study suggests that bean sprouts can be a superior substitute for lettuce, thus resolving problems in seasonal supply and cost fluctuations while improving productivity in mealworm farms. The study also shows that combining the vegetables with milk as well as increasing the amount of wet feed provided to the mealworms are additional means of enhancing larval growth rates.

**Abstract:**

The *Tenebrio molitor* larva (yellow mealworm) holds great potential as a sustainable ingredient in food and feed. Optimizing its growth under mass farming requires careful water management. However, the availability and cost of fresh fruit and vegetables, which are the most widely used sources of water, can vary geographically, which calls for the search for relatively affordable, effective, and readily available alternatives. We evaluated the effect of three water sources (bean sprouts, lettuce leaves, and milk) as well as their quantity on weights and nutrient profiles of reared *T. molitor* larvae. Newly hatched mealworm larvae were maintained in controlled conditions of 25 °C and 60% relative humidity under a 12-h light–dark cycle for 15 weeks. When provided as sole-supplements, bean sprouts induced the highest larval weight gains compared to fresh lettuce leaves, which in turn performed better than milk and water. However, the addition of milk to the vegetable supplements enhanced growth. Furthermore, doubling the level of water supply resulted in 70% higher larval weights by week 14 post hatching. Moreover, water sources did not change the nutrient content of the harvested larvae. These findings suggest that mealworm productivity can be enhanced by increasing water feed levels and that bean sprouts may be a superior alternative to lettuce.

## 1. Introduction

Insects have gained attention as an alternative to animal-sourced proteins for the food and feed industries in the recent two decades [1,2]. This has occurred amidst concerns over the unsustainability in food and feed production, especially as the global population, and thus, demand grows [3,4,5]. Insect production is lauded as being environmentally sustainable because it is associated with low greenhouse gas emission intensities, high feed conversion efficiency, limited land requirements, and low water requirements [2,6]. Recent estimates indicate that over 2100 insect species are traditionally consumed by over two billion people across the globe [2,7]. From these, several insect species with high growth rates, high oviposition rates, tolerance of high stocking densities, and low susceptibility to disease among other traits have been identified as candidates for industrialization.

*Tenebrio molitor,* a member of the order Coleoptera (beetles), is one of the insects being promoted for industrialization of entomofarming [2,8]. The larval forms of *T. molitor* (yellow mealworms) are commonly reared for niche markets (food for pets and zoo animals) as well as for aquaculture and livestock (pigs and poultry) feed in parts of Europe, Asia, and North America [9,10]. In addition to being used in animal feed, mealworms are permitted as an ingredient in food intended for human consumption in many parts of the world, including the European Union [11]. However, a few hurdles to the industrialization of mealworms and other edible insects still exist. Some of the priority areas to address include the optimization and automation of rearing conditions to improve competitiveness as well as regulatory frameworks to improve consumer confidence [2]. Also, given the relative plasticity of the mealworms’ growth requirements, further evaluation of nutritional inputs is required to optimize production [8].

An important factor in mealworm rearing is the management of water requirements. Generally, mealworms have very low water requirements; however, optimal growth is achieved with the calculated provision of a water source [12,13]. Overall, water requirements are satisfied by maintaining high relative humidities (>70% RH) within their environments as well as through the provision of a feed ingredient with high moisture content on top of their dry ration [8]. Excess moisture within their environment risks mold contamination, while pools of water are often a drowning risk. For these reasons, fruits and vegetables such as carrots and lettuce are commonly used as a water source due to their low dry matter content [8]. However, the availability and cost of fruits and vegetables can vary geographically. For example, in the Republic of Korea, the conventional water sources in mealworm farms are leafy vegetables such as lettuce, napa cabbage, or white radish [14]. These vegetables are seasonal, leading to volatility in their availability and prices, hence the need for an affordable, effective, and readily available alternative.

The objective of this research was to explore the use of bean sprouts and milk as alternatives to seasonal leafy vegetables as water sources for mealworms with the aim of enabling sustained production efficiency for mass-rearing insect farms. The suitability of alternative water sources was evaluated based on the weights and nutrient profiles of the reared *T. molitor* larvae.

## 2. Materials and Methods

### 2.1. Insects, Housing, Main Fodder, and Water Sources

*T. molitor* used in this study was kindly provided by a mass-rearing mealworm farm (HKnong, Namwon city, Jeollabuk-do, Republic of Korea). To initiate mating and oviposition, approximately 2000 adult beetles were placed in an oviposition tray inside a plastic rearing tray (60 × 40 × 15 cm) filled with 1 cm of wheat bran feed. After three days, oviposition was terminated, and the adult beetles were transferred to another oviposition tray. The larval-rearing facility was maintained at a temperature of 25 °C with a relative humidity of 60% under a 12-h light–dark cycle. Two kilograms of wheat bran were added to the rearing tray as the main fodder, and newly hatched larvae were reared in the tray for a period of 15 weeks. The trays were stacked 15 cm above each other to allow for aeration. Depending on feed consumption, an additional 1.5–2 kg of wheat bran were added subsequently.

The nutrient composition of the main feed (which was wheat bran) and various water sources, including lettuce, bean sprouts, and milk, were analyzed by the Korea Feed Ingredients Association (Sejong, Republic of Korea) using AOAC methods [15]. The results of this analysis are presented in Table 1.

### 2.2. Experimental Design and Procedures

Experiment 1 was conducted to evaluate prospective water sources that could replace leafy vegetables such as lettuce and napa cabbage. The water sources included whole milk and other vegetables, which were purchased from a local market. Prior to their inclusion, the vegetables were washed under tap water and air dried to minimize possible contaminants such as pesticides. The water treatments were arranged in a 2 × 2 × 2 factorial design to study two vegetable-based water sources (lettuce and bean sprouts) at two water supply levels (low and high) and with and without milk. Twenty-four larval groups were arranged into eight water treatments (three groups each), as summarized in Table 2. Each group was estimated to contain 5000 larvae (range = 4000–6000) reared in a 60 cm × 40 cm × 15 cm tray.

The water sources were introduced 20 to 30 days after hatching. For the larval groups that were reared with low levels of water, seven grams of the respective water sources were initially provided. The quantity of the water source was increased by 15 g in the subsequent water rations, which were provided twice a week, until 60 to 70 days post hatching, at which point the amount of water source material was fixed at 140 g per serving (280 g per week), until harvest. On the other hand, groups that were raised in high water levels were provided with twice as much water source material during the rearing period as those raised in low water levels.

To evaluate the effect of each water treatment on growth performance, we measured the fresh weight of larvae at intervals of 10 to 14 days until pupation occurred. One hundred larvae were randomly selected and weighed from each tray using a calibrated FX-300 electronic balance (A&D company Ltd., Tokyo, Japan), which had a precision of 0.001 g. Larval weights were computed as averages of the 100 larvae. Photographs of *T. molitor* larvae with various water sources during experiment period are shown in Figure 1.

Experiment 2 was conducted to determine whether nutrient composition of the produced larvae was influenced by the water source (lettuce, bean sprouts, and milk/lettuce combination). All larval groups were provided with wheat bran and equal amounts of the respective water source under test. The three test groups were replicated in six batches, and newly hatched larvae were reared in the tray for a period of 15 weeks, as described in experiment 1.

### 2.3. Analysis of Larval Nutrient Composition

Before harvest at the end of experiment 2, *T. molitor* larvae were starved for 72 h to empty their intestines and then washed with water. The larvae were freeze-dried and stored at −20 °C until analysis. Overall, the composition of crude protein, crude fat, crude ash, gross energy, sodium, potassium, and various essential and non-essential amino acids in larvae, were analyzed by the Korea Feed Ingredients Association (Sejong, Republic of Korea) following methods published by the Association of Official Analytical Chemists [15].

### 2.4. Statistical Analysis

A three-way analysis of variance (ANOVA) was used to analyze the main effects of water supply level (high and low), water source (lettuce and bean sprouts), milk inclusion, and their two-way and three-way interactions for larval weight data using Statistical Analysis Systems (SAS) for Windows, v9.1 (SAS Institute, Cary, NC, USA). Duncan’s multiple range test was conducted to evaluate pairwise differences between treatments within weeks. The rearing tray was considered an experimental unit, and all treatments were done in triplicate. Differences were considered statistically significant at *p* < 0.05.

## 3. Results

### 3.1. Effects of Milk and Lettuce as Water Sources on Larval Growth Rates

Initially, we compared lettuce, a conventional water source, to whole milk, or water as alternative moisture sources with respect to their effect on *T. molitor* larval growth (see Appendix A). Equal amounts of milk, water, and lettuce were provided to the larvae. Milk and water were first mixed with half the amount of wheat bran and then added. On day 92 after hatching, when the lettuce-fed larvae weighed 105.3 mg, the milk- and water-fed larvae weighed only 94 mg and 61.3 mg, respectively, indicating that water per se is not sufficient for the larval growth compared to leafy vegetables as a water source (Appendix A).

### 3.2. Larval Growth Depending on Water Supply Levels and Sources with or without Milk—Main Effects

For the main water source experiment, bean sprouts were included, which can be grown year-round everywhere in Republic of Korea and thus have a very stable supply and price. To determine the synergistic effect of milk in combination with vegetables, half of the amounts of vegetables were replaced by whole milk. Because *T. molitor* larvae are known to be voracious, the effect of amount levels of the water source was also examined. The results of the analysis of the larval weights among the eight study groups throughout the experiment are presented in Table 3.

The data show that high water levels started to increase the larval weight by 16% (from 13.00 mg to 15.08 mg, *p <* 0.0001) on day 46. By day 56, bean-sprout-supplemented larvae outweighed the lettuce-supplemented larvae by 17% (32.84 mg vs. 28.00 mg). Also, by day 56, milk inclusion caused 22% higher larval weights relative to the respective groups without milk (33.42 mg vs. 27.42 mg *p <* 0.0001). A comparison of the treatments on day 97 is presented in Table 3. At 97 days post hatching, larval weights were 70% higher among groups fed on high water levels compared to those on low water level supplies (117.00 mg vs. 199.17 mg; *p <* 0.0001). On the same day, the bean-sprout-supplemented groups had 14% higher weights than the lettuce-supplemented groups (147.67 mg vs. 168.50 mg, *p <* 0.0001), and groups with milk added to the vegetable supplements had 25% higher weights compared to those without milk (175.50 mg vs. 140.67 mg, *p <* 0.0001). The water source experiment and weight measurement continued until day 108. The experiment was terminated at day 108 following the observation of the first pupae (at day 101) among the groups that were provided with high water levels with milk, i.e., groups BM-2 and LM-2 (Figure 2). The final larval weights at day 108 were 54% higher in the groups with high water supply levels compared to corresponding groups with low water supply levels (217.08 mg vs. 141.25 mg, *p* < 0.0001) (Table 3). Likewise, the bean sprout groups had 12% higher larval weights compared to the lettuce groups (189.50 mg vs. 168.84 mg *p* < 0.0001), while milk-vegetable supplemented groups showed 10% higher larval weights compared to the corresponding groups with only vegetables (187.33 mg vs. 171.00 mg, *p* < 0.0001).

### 3.3. Larval Growth Depending on Water Supply Levels and Sources with or without Milk—Interactions

Two-way interactions among water levels, sources, and milk inclusion for larval growth were analyzed. For ‘water level—water source’ interactions, four categories were evaluated: bean sprouts at high water levels (B-2 & BM-2), bean sprouts at low water levels (B-1 & BM-1), lettuce at high water levels (L-2 & LM-2), and lettuce at low water levels (L-1 & LM-1). There were significant ‘water level × water source’ interactions for larval growth from day 67 to day 108 (*p* < 0.001, Table 3). Both lettuce- and bean-sprouts-supplemented groups had improved larval growth when provided with high water levels. However, at day 97, the relative increase in larval weight was higher among the bean sprouts groups (81.62% increase, that is, 119.67 mg vs. 217.34 mg in low and high-water levels, respectively) than in lettuce group (58.31% increase; i.e., 114.3 mg vs. 181.0 mg in low and high water levels, respectively) (*p* < 0.0001, Figure 3A). Groups in which bean sprouts were provided at high water levels (B-2 & BM-2) had the highest larval weights on day 97 for the ‘water level–water source’ categories, whereas larval groups that were provided with lettuce at low water levels (L-1 & LM-1) had the lowest weights.

To evaluate the water source—milk interactions among the groups, four categories were considered. These included bean sprouts with milk (BM-1 & BM-2), bean sprouts without milk (B-1 & B-2), lettuce with milk (LM-1 & LM-2), and lettuce without milk (L-1 & L-2). There were significant ‘water source × milk’ interactions from day 77 to day 97 (*p <* 0.01, Table 3). The effect of milk inclusion on larval growth was greater in lettuce (35.03%, i.e., 125.65 mg among groups without milk vs 169.67 mg among groups with milk) than in the bean sprout treatment groups (16.49%, i.e., 155.66 mg among groups without milk vs 181.34 mg among groups with milk) on day 97 (*p =* 0.0004, Figure 3B), implying that a limiting growth requirement in lettuce was alleviated by inclusion of milk. Larval groups allowed a 50:50 combination of bean sprouts and milk had the highest larval weights at day 97 compared to the other treatments, while the larvae groups that were fed lettuce without milk had the lowest weights.

Assessment of the milk–water level interactions considered the following four categories. High water levels including milk (BM-2 & LM-2), high water levels without milk (B-2 & L-2), low water levels including milk (BM-1 & LM-1), and low water levels without milk (B-1 & L-1). Significant ‘milk × water level’ interactions were detected from day 56 to day 108 (*p <* 0.01, Table 3). By day 97, the effect of milk inclusion on larval growth was greater in high water level groups (25.05% difference, i.e., 221.34 mg in groups with milk vs 177 mg in groups without milk) than in low water level groups (24.28% difference, i.e., 129.67 mg vs 104.33 mg in groups with and without milk, respectively) (*p =* 0.0003, Figure 3C). Larval groups with high water levels and including milk had the highest average larval weights at day 97, whereas larval groups that were provided with low water levels without milk had the lowest larval weights. In both with- and without-milk inclusion treatments, high water levels improved larval growth by over 70% compared to the low water level groups. Based on these interactions, the high-water level feed of bean sprouts with milk inclusion had the highest larval weight on day 97, followed by high water level lettuce with milk inclusion. These high-water level feeds without milk inclusion were the next most efficient treatments for larval growth, followed by low water level feeds. Among the low-level water supply tests, a combination of water feeds of milk and vegetables resulted in a 20 to 30% increase in larval weight compared to vegetable feeds without milk inclusion.

### 3.4. Evaluation of Nutrient Composition of Larvae Fed with Different Water Sources

The nutrient compositions of wheat bran and water sources are shown in Table 1. Although the nutrient contents, such as gross energy, protein, and lipids, between the three water sources are different, it is not clear at this point which nutrient contents affected the larval growth rate fed with different types of water sources. Initially, because the overall amount of water source material during the entire rearing period was approximately half of the total main fodder, it was predicted that there would be differences in the nutritional composition of larvae fed with three different water sources, but no significant difference was found in the nutritional analysis (Table 4). Thus, we can conclude that all three types of water source material affect the larval growth to a similar extent without any significant changes in the nutritional composition of the larvae. According to this result, it is expected that water source material, such as bean sprouts or milk–vegetable combinations, can provide a better solution in the supply of water sources, which has been difficult due to the seasonal supply and cost.

## 4. Discussion

The mealworm is known to thrive without water in stored, dry grain and flour and can therefore be farmed without provision of water [12,16]. This is because the larvae are capable of satisfying their water requirements with the metabolic water derived from digestion of grain or whole meal flour [12]. However, it was observed early on that larvae are drawn to moist parts of their feed and they do attain higher growth rates when allowed access to such water sources [17,18]. Further, when mealworms’ diet is based largely on wheat bran (commonly provided in mealworm farms and as used in this study), a water source is required since the bran is high in fiber and thus limited in metabolic water content. Various vegetable-based water sources are used in mealworm farming, and these could have varying effects due to their variability in moisture, nutrients, and metabolite composition [19,20]. Here, we set out to evaluate the use of lettuce, bean sprouts, and milk as moisture sources in mealworm rearing regarding their impact on larval weight gain and nutrient composition. This work demonstrates the importance of additional supplies of water sources as well as the ability of bean sprouts to replace lettuce as a moisture source in mealworm rearing.

Our results indicate that fresh lettuce, as a water source, has a higher positive influence on larval growth rate compared to either water or milk when provided alone. It is a common practice in many Korean mealworm farms and elsewhere in America and Europe to provide mealworms with water by spraying into each tray (rearing unit) [21]. The comparative advantage of fresh lettuce leaves over plain water is unsurprising considering the supply of additional nutrients (to the wheat bran basal diet) that are availed in the lettuce. Mealworms have been shown to achieve enhanced growth rates when provided with protein-rich diets [22]. While milk has not been specifically studied as a feed supplement in mealworms, it does have the potential to supply enough moisture and protein to the growing larvae. The crude protein and moisture content within milk is quantitatively comparable to that in the conventionally used fresh lettuce leaves. A possible explanation for the faster growth observed among lettuce-supplemented larvae in this study over those provided with milk could lie in the relative composition of essential micronutrients availed in these two substrates. Indeed, fresh lettuce is rich in dietary minerals, vitamins, and bioactive compounds, which are a good addition to the nutrients in the wheat bran, thus playing a crucial role in larval growth [23].

Mealworm larvae grow even faster when supplemented with bean sprouts compared to when supplemented with fresh lettuce leaves. This may be a reflection of the higher nutrient content in the bean sprouts relative to the fresh lettuce leaves. With nearly twice the dry matter content and more than twice the content of gross energy, crude protein, and crude fat, the bean sprouts are a rich substrate for the mealworms’ growth. It has also been shown that bean sprouts contain high amounts of phytochemicals and vitamin C, which is important in the molting process [24,25,26,27].

The addition of milk to the vegetable substrate improves larval weight gain for both the lettuce- and bean-sprout-supplemented mealworm groups. This is consistent with findings from a study by Konala and colleagues, which showed that silkworms grew faster when milk was added to their diet of fresh mulberry leaves [28]. Interestingly, in our study, adding milk in a 1:1 ratio to either fresh lettuce leaves or bean sprouts tended to reduce the difference in effect on mealworm growth rate due to either vegetable availed singly. This indicates that milk supplies some key limiting nutrients that are critical to optimal larval growth. An example of a micronutrient in milk that is limited in the two vegetables is cholesterol, which plays important roles in insect physiology, for example, as a precursor for hormones and which insects obtain exogenously (in their diets) [29,30,31].

Overall, in this study, doubling the quantity of moist feed substrates caused up to 70% increments in larval weight by 97 days post hatching, thus shortening the growing periods by nearly 30 days. Ortiz and colleagues [21] suggest that an average ration of 0.006 g to 0.010 g of fresh vegetables be provided per larva per day. In this study, our categories of low and high-water levels correspond to the low and high end of this range with a slight adjustment (i.e., 0.006 g to 0.012 g per larva per day). Currently, the amounts of fresh vegetables provided in the still few and emerging commercial farm operations in Republic of Korea correspond to this lower end of the range. Worse still, many small-scale farms only provide a single fresh (lettuce) leaf per tray that is replenished twice a week or when the leaf is found dehydrated. The improvement in productivity demonstrated in this study in association with doubling the water supply is undoubtedly welcome for commercial mealworm farming. However, caution is required while increasing the moisture supply, as this comes with some new risks and challenges. For instance, increased moisture risks microbial and fungal accumulation, which would degrade the provided feed, increase larval mortality, and compromise the safety of the produced mealworm as a food or feed [11,21,32]. We also found that the tested moisture sources showed no differential effect on the nutrient composition of harvested larvae. Nonetheless, the possible influence on nutrient composition of the produced larvae needs continuous investigation.

## 5. Conclusions

Our work suggests that bean sprouts are a reliable water source with growth-enhancing effects in growing *T. molitor* larvae. The study builds on previous reports that established the growth-enhancing effect of vegetable supplements in mealworm farming [33,34,35]. Furthermore, findings from this work show that the enhanced growth rate due to supplementation with bean sprouts does not compromise the nutrient content of the harvested larvae. Finally, the work demonstrates that combining milk with vegetables and increasing wet feed volumes provided to the mealworms can boost larval growth rates.

This work presents opportunities for improving productivity of mealworm mass-rearing operations and offers alternatives that are essential to addressing challenges in seasonal supply and cost fluctuations of some vegetable water sources. Future studies should explore the effect of additional water feed on adult mortality, fecundity, or even pupal eclosion rate. Also, more research is needed to optimize the water supply based on various vegetables used in mealworm farming.

## Figures and Tables

**Figure 1 animals-14-00895-f001:**
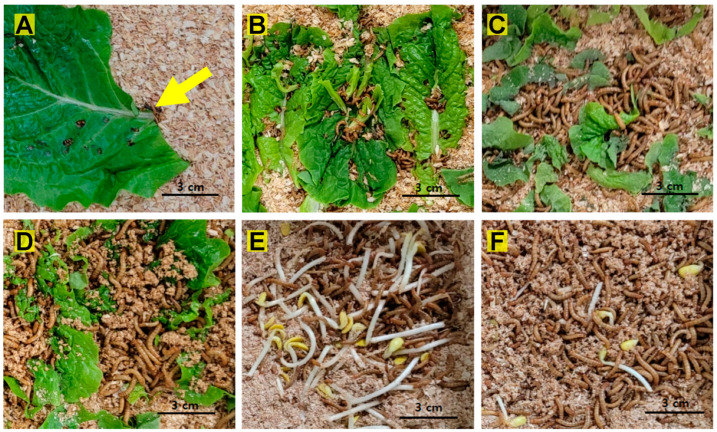
Photographs of *Tenebrio molitor* larvae with various water sources during experiment period. (**A**) Larvae (at the tip of the yellow arrow) with lettuce on day 40 after hatching. (**B**) Larvae with lettuce on day 60. (**C**) Larvae with lettuce on day 85. (**D**) Larvae with 50% lettuce + 50% milk on day 85. (**E**) Larvae with bean sprouts on day 85. (**F**) Larvae with 50% bean sprouts + 50% milk on day 85.

**Figure 2 animals-14-00895-f002:**
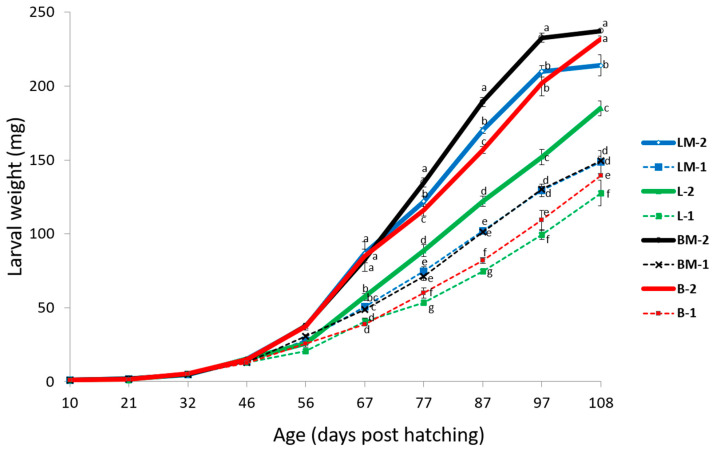
Larval growth curve of *Tenebrio molitor* fed with different water supply levels and sources. L-1, lettuce-fed larvae (control); B-1, bean sprouts-fed larvae; L-2, double the amount of L-1; B-2, double the amount of B-1; LM-1, 50% lettuce + 50% milk; BM-1, 50% bean sprouts + 50% milk; LM-2, double the amount of LM-1; BM-2, double the amount of BM-1. Symbols represent mean larval weight per group ± SD (n = 3). Significant differences were determined according to Duncan’s multiple range test and means with different superscripts are significantly different at *p <* 0.05.

**Figure 3 animals-14-00895-f003:**
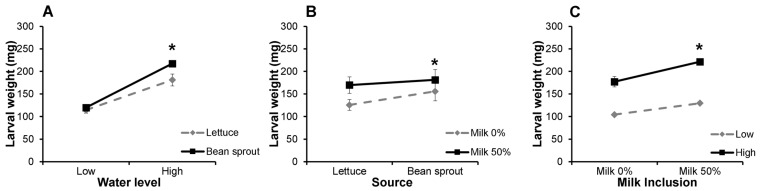
Two-way interactions between water levels and sources, milk inclusion and sources, and water levels and milk inclusion on day 97. Line graphs showing the (**A**) ‘water level × source’ interaction, (**B**) ‘source × milk’ interaction, and (**C**) ‘milk × water level’ interactions. These interactions were analyzed using a three-way analysis of variance (ANOVA) (* *p* < 0.001). Symbols represent average weight per larva ± SEM (n = 6).

**Table 1 animals-14-00895-t001:** Nutrient composition of main fodder and water sources (as fed-basis).

Nutrients ^1^	Main Fodder	Water Sources		
	Wheat Bran	Lettuce	Bean Sprout	Milk
GE (kcal/kg)	4071	335	777	677
Dry matter (%)	87.31	8.79	14.24	11.89
Moisture (%)	12.69	91.21	85.76	88.11
Crude protein (%)	15.19	2.44	7.22	2.99
Crude fat (%)	3.90	0.56	2.38	2.35
Crude ash (%)	4.17	1.98	0.99	0.67
Crude fiber (%)	9.59	0.74	1.04	0.00
NFE (%)	54.46	3.07	2.61	5.87
Ca (%)	0.27	0.20	0.05	0.11
P (%)	2.40	0.04	0.12	0.08
Mg (%)	0.42	0.05	0.05	0.01
Fe (ppm)	134.12	10.55	9.17	1.20

^1^ All data on nutrient composition is based on analyzed values. GE, gross energy; NFE, nitrogen-free extract.

**Table 2 animals-14-00895-t002:** Summary of the water source treatments in the larval groups.

Treatment Group	Vegetable Water Source	Milk ^1^	Level ^2^
L-1	Lettuce	−	Low
B-1	Bean sprouts	−	Low
LM-1	Lettuce	+	Low
BM-1	Bean sprouts	+	Low
L-2	Lettuce	−	High
B-2	Bean sprouts	−	High
LM-2	Lettuce	+	High
BM-2	Bean sprouts	+	High

^1^ When provided (in groups LM-1, BM-1, LM-2, and BM-2), milk made up 50% of the water source materials by weight. On application, the milk was mixed with half of the wheat bran and provided in lumps. The “+” and “–“refer to presence or absence of milk, respectively. ^2^ Treatments B-2, LM-2, and BM-2 were allowed double the amount of water source material as in the corresponding groups (B-1, LM-1, and BM-1) that were provided low water levels.

**Table 3 animals-14-00895-t003:** Effect of water level (low and high), source (lettuce and bean sprouts), and milk inclusion (− and +) on growth of *T. molitor* larvae.

Age (Days Post Hatching)	Larval Weights (mg)						SEM	*p*-Value						
	Main Effects							Main Effects			Interactions			
	Level		Source		Milk			Level	Source	Milk	Level × Source	Source × Milk	Milk × Level	Level × Source × Milk
	Low	High	Lettuce	Bean Sprouts	−	+								
Day 10	1.00	1.00	1.00	1.00	1.00	1.00								
Day 21	1.42	1.84	1.59	1.67	1.42	1.84	0.208	0.0399	0.6607	0.0399	0.6607	0.6607	0.6607	0.6607
Day 32	4.92	5.00	4.92	5.00	5.17	4.75	0.550	0.8375	0.8375	0.3126	0.5404	0.8375	0.8375	0.8375
Day 46	13.00	15.08	14.25	13.84	14.00	14.08	0.281	<0.0001	0.0962	0.7283	0.7283	0.3046	0.7283	0.7283
Day 56	26.25	34.58	28.00	32.84	27.42	33.42	0.784	<0.0001	<0.0001	<0.0001	0.1220	0.7890	<0.0001	0.0027
Day 67	44.67	77.84	58.84	63.67	55.34	67.17	1.682	<0.0001	0.0089	<0.0001	0.0007	0.2757	0.0002	0.0002
Day 77	64.83	115.34	84.67	95.50	79.50	100.67	1.834	<0.0001	<0.0001	<0.0001	<0.0001	0.0024	0.0003	0.3981
Day 87	90.00	159.50	117.17	132.33	108.84	140.67	1.237	<0.0001	<0.0001	<0.0001	<0.0001	<0.0001	<0.0001	0.0657
Day 97	117.00	199.17	147.67	168.50	140.67	175.50	2.649	<0.0001	<0.0001	<0.0001	<0.0001	0.0003	0.0004	0.0431
Day 108	141.25	217.08	168.84	189.50	171.00	187.33	2.590	<0.0001	<0.0001	<0.0001	<0.0001	0.7064	0.0011	0.1644

A three-way analysis of variance (ANOVA) was used to analyze the main effects and their two-way and three-way interactions for larval weight. Data represent mean of larval weights in all replicates within the corresponding treatment groups. SEM, standard error of mean.

**Table 4 animals-14-00895-t004:** Nutrient composition of *T. molitor* larvae fed with different water sources.

Nutrients	L	B	LM	*p*-Value
Crude Protein (%)	54.85 ± 0.94	55.32 ± 1.32	55.03 ± 2.50	0.8949
Crude Fat (%)	33.57 ± 0.81	33.58 ± 1.60	34.10 ± 2.29	0.8256
Crude Ash (%)	4.21 ± 0.14	4.23 ± 0.10	4.37 ± 0.61	0.7121
GE (kcal/kg)	6672 ± 51	6744 ± 123	6811 ± 85	0.0567
Ca (%)	0.16 ± 0.06	0.17 ± 0.04	0.17 ± 0.04	0.8050
P (%)	0.79 ± 0.08	0.85 ± 0.05	0.80 ± 0.08	0.3331
Essential amino acids				
Arginine (%)	2.68 ± 0.15	2.73 ± 0.11	2.73 ± 0.08	0.7336
Histidine (%)	1.70 ± 0.10	1.71 ± 0.09	1.70 ± 0.07	0.9526
Isoleucine (%)	2.31 ± 0.14	2.37 ± 0.08	2.38 ± 0.10	0.4879
Leucine (%)	3.77 ± 0.19	3.85 ± 0.14	3.87 ± 0.16	0.5761
Lysine (%)	2.98 ± 0.26	3.15 ± 0.11	3.17 ± 0.08	0.1489
Methionine (%)	0.73 ± 0.06	0.78 ± 0.04	0.78 ± 0.06	0.2477
Phenylalanine (%)	1.79 ± 0.09	1.83 ± 0.06	1.88 ± 0.05	0.1229
Threonine (%)	1.89 ± 0.10	1.9 ± 0.08	1.89 ± 0.05	0.9065
Valine (%)	3.54 ± 0.19	3.58 ± 0.17	3.56 ± 0.12	0.9150
Non-essential amino acids				
Alanine (%)	4.42 ± 0.30	4.43 ± 0.29	4.49 ± 0.18	0.8960
Aspartic acid (%)	3.91 ± 0.20	3.98 ± 0.19	3.97 ± 0.21	0.8240
Cystine (%)	0.58 ± 0.06	0.62 ± 0.04	0.58 ± 0.07	0.3448
Glutamic acid (%)	5.43 ± 0.30	5.58 ± 0.20	5.62 ± 0.21	0.3767
Glycine (%)	2.81 ± 0.16	2.86 ± 0.14	2.86 ± 0.11	0.7738
Proline (%)	3.27 ± 0.23	3.50 ± 0.18	3.45 ± 0.11	0.1140
Serine (%)	1.96 ± 0.10	1.94 ± 0.11	1.90 ± 0.04	0.4718
Tyrosine (%)	3.10 ± 0.15	3.00 ± 0.22	3.01 ± 0.33	0.7418

Data represent mean ± SD (n = 6) on dry matter basis. All means are not significantly different according to Duncan’s multiple range test at *p <* 0.05. L, lettuce-fed larvae; B, bean sprouts-fed larvae; LM, lettuce-&-milk-fed larvae.

## Data Availability

The data that support the findings of this study are available within the manuscript and in the tables provided within the Appendix A alongside the paper.

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
