# Peer review of "Bean Sprouts, Lettuce, and Milk as Water Sources in Tenebrio molitor Larval Growth"

_animals, 2024, doi:10.3390/ani14060895_

Round 1

Reviewer 1 Report

Comments and Suggestions for Authors

The manuscript “Bean sprouts, lettuce, and milk as water sources in Tenebrio molitor larval growth" is interesting because it investigates an aspect that is poorly considered in the feeding of farmed tenebrio: the management of the water source. However, the effects of the different water source are evaluated only on larval weight and nutrient content, while the analysis of larval development times would have given further value to the manuscript. In Materials and Methods, a table of different treatments would help readers. In the results, check for some unclear statements. Conclusions should be improved and references completed according to journal guidelines.

Below are the specific comments:

SUMMARY

Line 18: please, delete “(and other seasonal leafy vegetables)”: the study does not demonstrate this.

INTRODUCTION

Line 42: please, replace “2” with “two”.

Line 55: integrate with use in aquaculture, having been the first authorised sector .

Line 56: reference [9] is not specific to the sentence.

MATERIALS AND METHODS

Line 106: please, replace “;” with “:”.

Line 106-119: a table/scheme would help the reader to focus on the treatments.

Line 123: please, delete “In the tables and figures,” .

Line 129: please, delete “fodder based on”.

Figure 1: Figure 1A is out of focus. If possible, replace with a better photo.

RESULTS

Line 156- 165: this is a preliminary test not described in M&M. It should be mentioned in M&M as justification for the choice to use milk and not water. If necessary, use the additional materials to describe the preliminary test and its results.

Line 173: please, replace “8” with “eight”.

Line 174-175: in this sentence it is superfluous to cite Figure 2 because the described results relate only to Table 2.

Line 175-176: please, check the structure of sentence.

Figure 3: this figure repeats the data shown in Table 2: delete

Line 188: the results in figure 2 are poorly valued. It would appear that the larvae in the high water level treatments are pupating, while the low level ones are still in the delayed growth phase. If the hypothesis is correct, is it possible to add data about it?

Tab 2: delete,  excess decimals (Day 46/Level High).

Line 224-225: check, sentence is not supported by the data in Table 2

Line 231: check, sentence is not supported by the data in Table 2

Line 252: for homogeneity, it is better to indicate the % increase (as in A and B).

DISCUSSION

Line 322-324: This is an important aspect and not sufficiently developed in the discussions. In Figure 2 it seems that some of the treatments are still growing. So the availability of water greatly influenced development times. The authors could comment and quantify whether the low level was sub-optimal and hypothesize an optimal water intake.

CONCLUSIONS

The conclusions should be better focused on the main results and possible developments.

DATA AVALAIBILITY STATEMENT

Please, why not applicable?

REFERENCES

Complete according to the journal guidelines.

Comments on the Quality of English Language

The English language requires moderate editing

Reviewer 2 Report

Comments and Suggestions for Authors

This paper presents results on T. molitor rearing under different diets. The study may of a certain practical interest. The research was adequately planned, and results are clearly presented. However, some aspects should be improved.

General comments

1. Provide details about how larvae were weighted. Instrument sensitivity? Di you measure fresh or dry weight? Were individual larvae weighted separately, or groups of larvae were weighted and then individual weight obtained by divided the total weight by the number of individuals?

2. Statistical analyses.  Provide full results (SS, df, F, p levels, etc.) for all ANOVAs. These new tables can be included as supplementary materials.

3. Improve resolution for Figure 1, 3, 4 (these figures are blurred)

4. Reference do not follow journal style. Revise.

Specific comments/language corrections

Line 13: revise grammar. “amin” is not appropriate here

14: yellow mealworm, a larva of one of the beetle species (Tenebrio molitor), is the most ->

 yellow mealworms (larvae of the beetle Tenebrio molitor) are the most

16:  rely on seasonal leafy vegetables ->  rely on leafy vegetables

24:  farming necessitates careful ->  farming needs careful

26:  geographically, necessitating the ->  geographically, which calls for

31:  weeks. Results show that when provided ->  weeks. When provided

33: I cannot understand likewise here

42:  in the recent 2 decades ->  in the last two decades

45: friendly –> sustainable

52 do not use italics for Coleoptera

56: Revise grammar and punctuation: I cannot understand this sentence

57: countries including, within the European union Do you mean alla EU or only some countries within EU? Revise, grammar is unclear

71: such as carrots, and lettuce -> such as carrots and lettuce

73-74: any reference to support this sentence?

106: The treatments included; L-1 ( -> The treatments included: L-1 (

175: “Figure 2 and the. Data shows”. This does not make sense. Please, revise.

180: presented in in Table 2 -> presented in Table 2

228: … replacing 50% of vegetable sources with milk resulted…. This part is unclear, please revise

234: … replacing 50% 234 of vegetable sources with milk… This part is unclear, please revise

263-4: Thus, it was concluded that all three types of water source material could affect the larval growth to a -> Thus, we can conclude that all three types of water source material affect the larval growth to a

270: There is no significant result in this table. Thus the sentence “means without superscript letters are not significantly different …” is not appropriate. I suggest: “row. All means are not significantly different …”

286: dramatic effects -> the importance of

Comments on the Quality of English Language

Language is acceptable. A few sentence need revision (see my specific comments)
